

# A systematic review on artificial intelligence techniques for detecting thyroid diseases

Lerina Aversano[1], Mario Luca Bernardi[1], Marta Cimitile[2], Andrea Maiellaro[1] and Riccardo Pecori[3,4]

[1] Department of Engineering, University of Sannio, Benevento, Italy
[2] Dept. of Law and Digital Society, UnitelmaSapienza University, Rome, Italy
[3] Institute of Materials for Electronics and Magnetism, National Research Council, Parma, Italy
[4] SMARTEST Research Centre, eCampus University, Novedrate (CO), Italy

## ABSTRACT

The use of artificial intelligence approaches in health-care systems has grown rapidly over the last few years. In this context, early detection of diseases is the most common area of application. In this scenario, thyroid diseases are an example of illnesses that can be effectively faced if discovered quite early. Detecting thyroid diseases is crucial in order to treat patients effectively and promptly, by saving lives and reducing healthcare costs. This work aims at systematically reviewing and analyzing the literature on various artificial intelligence-related techniques applied to the detection and identification of various diseases related to the thyroid gland. The contributions we reviewed are classified according to different viewpoints and taxonomies in order to highlight pros and cons of the most recent research in the field. After a careful selection process, we selected and reviewed 72 papers, analyzing them according to three main research questions, *i.e.*, which diseases of the thyroid gland are detected by different artificial intelligence techniques, which datasets are used to perform the aforementioned detection, and what types of data are used to perform the detection. The review demonstrates that the majority of the considered papers deal with supervised methods to detect hypo- and hyperthyroidism. The average accuracy of detection is high (96.84%), but the usage of private and outdated datasets with a majority of clinical data is very common. Finally, we discuss the outcomes of the systematic review, pointing out advantages, disadvantages, and future developments in the application of artificial intelligence for thyroid diseases detection.

# INTRODUCTION

The thyroid is an important human endocrine organ, positioned in the anterior part of the neck, and it secretes the hormone which regulates the human metabolism. A thyroid disorder takes place whenever this organ produces either too many or too few hormones. Thyroid disorder is the most common disease in the endocrine field (*Longbottom & Macnab, 2014*), causing several ailments and, in its more severe forms, also death. With respect to several other diseases, thyroid diseases affect about 200 million people worldwide.

Corresponding author
Riccardo Pecori,
riccardo.pecori@uniecampus.it

Moreover, an estimated 40 percent of the world population is at risk of iodine nutrient deficiency, essential for the production of thyroid hormones, that is the major cause of morbidity worldwide (*Keestra, Högqvist Tabor & Alvergne, 2020*; *Aversano et al., 2021a*). For this reason, it is very important to diagnose thyroid diseases at their early stages and take precautions to avoid the most dangerous conditions. There are several approaches to diagnose thyroid disorder such as clinical test evaluation, imaging inspection, blood analysis, and tissue biopsy. These approaches require the work of doctors and are not foolproof because the process of diagnosis of thyroid disorder from the laboratory analysis is quite complex and requires the doctors' extensive knowledge and experience. For this reason, in the last years, several research studies, using Artificial Intelligence (AI) techniques to detect specifically various thyroid diseases (*Ma et al., 2019*; *Kwak & Hui, 2019*), have been carried out. Indeed, AI has been extensively used to solve problems in the healthcare field (*Kwak & Hui, 2019*; *Aversano et al., 2021b*) and has often provided good accuracy. This is further spurred by the increasing computational power of computers, which allow running the most complex and time-consuming AI algorithms.

Despite the increasing number of studies about AI to predict thyroid diseases, a broader discussion and comparison of the used AI approaches in the given context is lacking. In this direction, the proposed systematic review aims to provide some useful insights for all the clinicians and researchers focusing on the topic of thyroid disease detection by identifying the main adopted AI techniques, the most adopted datasets, and the types of data useful to perform thyroid disease detection.

The target audience for this review includes computer scientists, bioinformatics specialists, data analysts, as well as medical doctors and endocrinologists in particular.

The main objectives of this systematic review are summarized in the following:

- summarizing the most recent AI solutions linked to the early detection of thyroid diseases (Research Question RQ1);
- identifying the used datasets to apply AI solutions for the early detection of thyroid diseases (Research Question RQ2);
- summarizing the most used data types to detect thyroid diseases using AI techniques (Research Question RQ3).

The rest of the article has the following structure. In section 'Background', concepts useful to understand the proposed investigation are reported. In section 'Related Work', we summarize, from a critical point of view, some recent reviews on the topic, obtained in the analysis and filtering process. In section 'Research Method', the research method adopted in this systematic review is described, together with the research questions, the databases used, the keywords, and the filtering as well as the inclusion/exclusion criteria. In section 'Results', we summarize and classify the considered papers, in a way suitable to answer the research questions. Finally, section 'Discussion' reports a discussion of the obtained findings, also highlighting the current research gaps, while section 'Conclusions' concludes the article with an overview of the systematic review and of the obtained results, drawing also some possible future developments in the field.

## BACKGROUND

The primary function of the thyroid gland is the production of the triiodothyronine (T3) and thyroxine (T4) hormones. These hormones travel through the body and help the regulation of the metabolism, while also aiding brain development, digestive function, muscle control, and mood balancing. Autoimmune diseases and nutrient deficiencies are the principal causes of thyroid complications (*Monaco, 2003*). Thyroid dysfunction is rather common in the general population, and mild or sub-clinical forms can be present in more than 10% of individuals older than 80 years. Diagnosing thyroid disease in old individuals can be difficult due to the non-specific clinical presentation of thyroid dysfunction and the impact of aging on the test results.

There are different kinds of thyroid dysfunction, namely, goiter, hyperthyroidism, hypothyroidism, malignant thyroid nodules, thyroiditis, *etc.* They are briefly summarized in the following (*Monaco, 2003*)

- Goiter is a non-cancerous enlargement of the thyroid gland. The most common cause of goiter worldwide is iodine deficiency in the diet;
- Hyperthyroidism occurs when the thyroid gland is overactive. It produces too much of its hormone;
- Hypothyroidism is the opposite of hyperthyroidism. The thyroid gland is underactive, and it cannot produce enough of its hormones;
- Thyroid nodules are growths that form on or in the thyroid gland. The nodules can be solid or fluid-filled, most are benign, but they can also be cancerous in a small percentage of cases;
- Thyroiditis can be considered a swelling of the thyroid.

Another pathology is Euthyroid, a normal thyroid hormonal functional state, but involved in initial structural changes such as goiter, cold nodule, multiple nodule goiter (MNG), and cancer (Grave's Disease and the like).

In the rest of the article, we will specifically focus on the following general and more frequent thyroid-related diseases: hypo- and hyperthyroidism, thyroid cancer, and euthyroid sick state. Other thyroid-related diseases will be considered, but in a general thyroid disease category.

## RELATED WORK

In this section, we summarize the existing reviews and surveys on the studied topic and we highlight the differences, compared with our proposed literature review. First of all, it is important to specify that in the literature there are currently very few systematic reviews or surveys about the use of artificial intelligence detection techniques for thyroid dysfunction.

Some studies discuss a more general context regarding both clinical and computerized thyroid dysfunction detection techniques. *Parmar & Mehta (2020)*, for example, discuss different kinds of dysfunctions which affect the thyroid and point out the main methods and processes used currently for detecting these dysfunctions from a clinical perspective. Moreover, they illustrate the computer-supported detection techniques distinguishing

them on the basis of the used input modes. The article also outlines the main strengths and open research problems to be addressed in this research area. A set of parameters, such as the feature extraction method and classification approach, is used to compare the discussed techniques. A general survey is also proposed by *Razia, Siva Kumar & Rao (2020)* reporting an overview of various machine learning techniques in medicine.

Alternative contributions are generally focused on one particular diagnostic method (*Chen, You & Li, 2020*), thyroid disease (*Abdolali et al., 2020*; *Chen, You & Li, 2020*; *Garg & Mago, 2021*) or one specific task (*i.e.*, segmentation, classification, diagnosis) (*Ludwig et al., 2023*). For example, *Chen, You & Li (2020)* propose a review and categorization of thyroid gland and thyroid nodule segmentation methods according to their own theoretical bases. In particular, the review compared 28 representative papers selected in the literature and found out that the most common methods for thyroid disease detection are based on machine and deep learning techniques. Moreover, the study found that the usage of big data for training provide better segmentation performance and robustness. However, deep learning models usually require large training datasets and imply a long training time. For thyroid nodule segmentation, the most commonly adopted methods are contour and shape-based methods, which lead to satisfactory performance results. Nevertheless, they are often tested on small datasets.

Similarly, in *Abdolali et al.(2020)* and in *Ludwig et al. (2023)*, the authors provide a systematic review of artificial intelligence applications focused on thyroid cancer diagnosis. *Abdolali et al. (2020)* considered and classified more than 50 papers discussing approaches for thyroid cancer detection exploiting AI algorithms. The paper also proposed future trends and challenges in the field and perspectives of computer-aided analysis to improve the efficiency of future methods for thyroid cancer diagnosis. In *Ludwig et al. (2023)* and *Xue et al. (2022)* the particular focus is on the usefulness of AI in ultrasonography for the diagnosis and characterization of thyroid cancer. *Sorrenti et al. (2022)* also report an overview of the state of the art regarding AI implementation for thyroid nodule ultrasound characterization and cancer. These recent studies show an increasing interest in the last year about the adoption of AI in the thyroid nodule ultrasound context. Finally, *Mendoza & Hernandez (2021)* provide a short review assessing and analyzing existing data mining methods for diagnosing thyroid diseases. This study, in the limited room of a conference, does not provide any keywords or research questions, thus does not adhere to Barbara Kitchenham's guidelines (*Kitchenham, 2004*) for a systematic review either.

There are also recent reviews that have a specific focus on a certain type of data. Considering AI in the context of improving thyroid health, some works focus only on images, such as *Bini et al. (2021)* and *Sharifi et al. (2021)*, and do not consider clinical data at all.

Differently from all the aforementioned papers, the proposed systematic literature review aims to perform a comprehensive investigation on the use of artificial intelligence approaches for the diagnosis and classification of the main thyroid dysfunctions, providing the mapping of the considered approaches with the used data types and the available datasets. We have considered different data types, like images and clinical data, and we have included all main thyroid dysfunctions. We have filtered out works tackling thyroid

diseases only as an example of the application of machine learning techniques; moreover, we have carefully adhered to Barbara Kitchenham's guidelines and the PRISMA selection procedure (*Page et al., 2021*) for a systematic review, clearly pointing out the investigated research questions, the formal analysis and filtering process we have followed, as well as the final results and discussion.

# RESEARCH METHOD

In order to properly conduct the literature review, we mainly adopted the guidelines proposed by *Kitchenham (2004)* and the PRISMA selection procedure (*Page et al., 2021*) for a systematic review. The phases performed are described in the following subsections.

## Research Questions and relevant keywords

The research questions we used to investigate the application of AI techniques to the detection of thyroid diseases are the following:

- RQ1: What are the main AI techniques used for the detection and identification of the most relevant thyroid diseases?
- RQ2: What datasets about thyroid diseases are used in the considered AI solutions?
- RQ3: What data types are used to detect and classify thyroid diseases using the considered AI techniques?

The first research question aims to discriminate the various thyroid diseases tackled by different AI-based detection approaches as well as their performance metrics.

AI field is rapidly advancing comprising a wide range of approaches and techniques. To understand the different ways in which AI is being used and developed for thyroid disease treatment, we have adopted a taxonomy of AI approaches, shown in Fig. 1, we used to classify and organize the various methods and applications of AI we found. This provides a framework for understanding the field and its potentials in the context of our systematic review. The selected categories are derived from *Witten, Frank & Hall (2011)* and are roughly based on the main categories of AI techniques (expert systems based on rule mining, decision tree-based or probabilistic-based machine learning models, artificial neural networks and hybrid/composite approaches):

1. **Probabilistic approaches**, which classify or group samples according to a certain probability distribution function;
2. **Kernel-based approaches**, which perform pattern analysis by transforming linearly inseparable data to linearly separable ones;
3. **Techniques using decision trees**, a model often used in operational research exploiting a tree-like structure of decisions on features, each representing a node of the tree itself;
4. **Rule-based approaches**, which determines interpretable classification strategies by means of relational rules made of certain antecedents and certain consequents;
5. **Neural networks**, which are models inspired by human brain made of layers of artificial perceptrons, connected in different ways, and useful for both classification and description tasks;

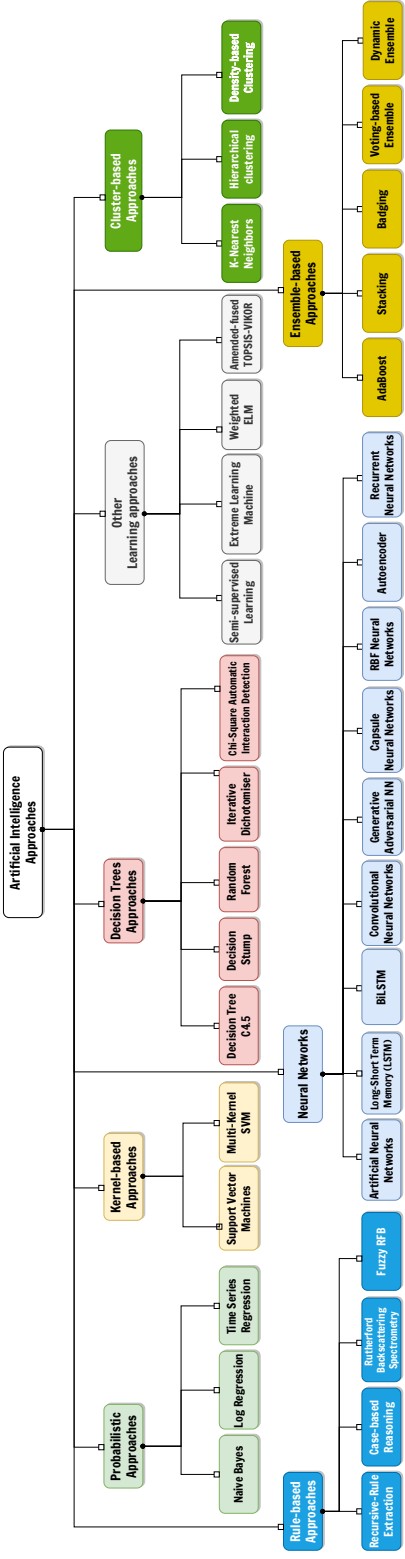

**Figure 1   The considered taxonomy of Artificial Intelligence approaches for the classification of the surveyed papers.**

6. **Cluster-based approaches**, based on the division of the group of samples according to a certain similarity metric;

7. **Ensemble approaches**, which combine different ML approaches and usually provides an output according to a certain strategy (*e.g.*, majority voting);

8. Other techniques or **hybrid methodologies**.

Conversely, for what concerns the thyroid disease, we focused on the following classification:

1. **hypothyroidism**, a situation of underactive thyroid gland when it does not produce enough of its crucial hormones;

2. **hyperthyroidism**, which is the opposite of hypothyroidism, is the situation when the thyroid gland produces too many of its hormones;

3. **euthyroid disease**, which can cause abnormal findings on thyroid function tests occurring in absence of any thyroidal illness;

4. **cancer** and malignant nodules;

5. **other thyroid-related issues** or general dysfunctions.

With reference to RQ2, we have analyzed the characteristics of the datasets used in the research studies, as well as their nature (private or public); whereas as regards RQ3 we have investigated whether the datasets were composed of images or clinical data, as well as the detailed list of the considered features. In this study, we have considered as clinical data all the structured data describing the state of health of the patients as well as their sociodemographic characteristics. These data are usually managed and analyzed in a different manner with respect to the medical images and other types of data (*i.e.,* natural language text).

We have converted the research questions into proper queries, used to search certain databases described in the following. The used queries (Q) are the following, wherein query 2 is used for answering both research question 2 and 3:

1. Q1: (''artificial intelligence'' OR ''machine learning'' OR ''deep learning'' OR ''neural network'' OR ''neural networks'') AND (''thyroid disease'' OR ''thyroid diseases'');

2. Q2: (''artificial intelligence'' OR ''machine learning'' OR ''deep learning'' OR ''neural network'' OR ''neural networks'') AND (''thyroid disease'' OR ''thyroid diseases'') AND ''dataset'';

## Searched databases

The selected papers were found in the following four main databases:

1. the database of the Institute of Electrical and Electronics Engineers (IEEE) (https://ieeexplore.ieee.org/Xplore/home.jsp), also called IEEEXplore, which contains technical articles in electrical engineering, electronics, computer science, and other related fields;

2. the Elsevier database (https://www.sciencedirect.com/), also called ScienceDirect, which permits to access to journals and technical and science papers in several scientific areas and fields;

3. the Springer database (https://link.springer.com/), also called SpringerLink, which permits one to access publications by the Springer Nature editorial group;

**Table 1  Criteria for including or excluding papers in the performed search process.**

| Acronym | Description of the criterium |
|---|---|
| *Inclusion criteria* | |
| $IC_1$ | Studies published in the year range 2016-2022 |
| $IC_2$ | Studies written in English |
| $IC_3$ | Studies should use AI techniques for analyzing any thyroid disease |
| *Exclusion criteria* | |
| $EC_1$ | The paper is a survey or review |
| $EC_2$ | The research does not consider artificial intelligence |
| $EC_3$ | The paper is not specifically focused on thyroid |

4. the database maintained by the Association for Computing Machinery (https://dl.acm.org/), also called ACM Digital Library.

Moreover, we have also searched the PeerJ CS database (https://peerj.com/computer-science/), to verify whether the considered topic was already regarded in the intended publication venue.

## Search process and filtering criteria

We have performed the search and filtering process represented in the PRISMA diagram in Fig. 2, wherein the inclusion ($IC$) and exclusion criteria ($EC$) presented in Table 1 were applied.

The selected papers fall in the January 2016–July 2022 range ($IC_1$). We have chosen this range in order to analyze only the most recent studies, dating back to at most 6 years ago. After applying the queries to the aforementioned databases we got a total of 375 papers, 35 from IEEEXplore, 220 from ScienceDirect, 102 from SpringerLink, and 17 from the ACM Digital Library. We have also found one paper in the PeerJ CS database.

Then, we skimmed titles and abstracts in order to remove possible duplicated as well as papers that (i) do not deal mainly with AI-based methods ($EC_2$), (ii) do not consider the thyroid gland and its diseases as the main focus ($EC_3$).

After this phase, a total of 106 papers remained. Finally, these remaining papers were retrieved and fully read for both extracting useful statistical information presented in the following and for filtering them further on the base of the following selection criteria:

- written in English ($IC_2$);
- specifically focused on thyroid ($EC_3$);
- an AI-based technique must be the core method of the paper to identify or classify thyroid diseases ($IC_3$);
- not a survey nor a review ($EC_1$).

This final filtering step resulted in a total of 72 papers to analyze, after removing also 12 recent surveys, some thereof already discussed in Section 'Related Work'.

Figure 3 presents the distribution over the years of the 72 papers selected for the systematic review. As one can see, there is a constant growth of the research on the topic over the recent years with the obvious exception of 2022, given that it can only consider

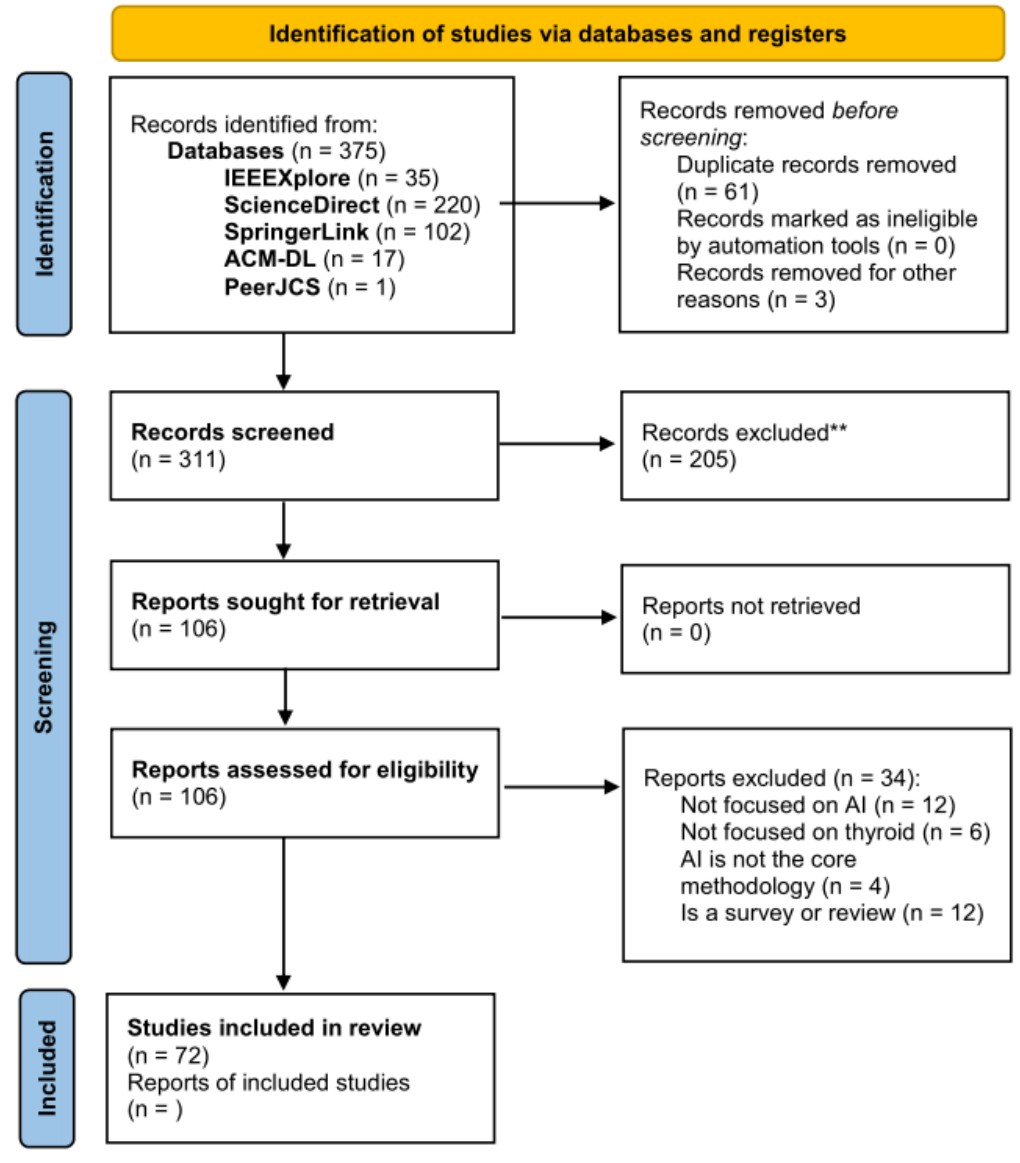

**Figure 2** PRISMA diagram of the selection procedure for the analyzed papers.

half of the year. The trend of the graph demonstrates an increasing interest in the usage of AI-based methods to detect and identify thyroid-related dysfunctions, thus corroborating our initial idea to research on this hot topic.

## RESULTS

In this section, we discuss the main results and outcomes of the systematic literature review, by following the research questions described in Section 'Research Method'.

## Distribution of papers per publication year

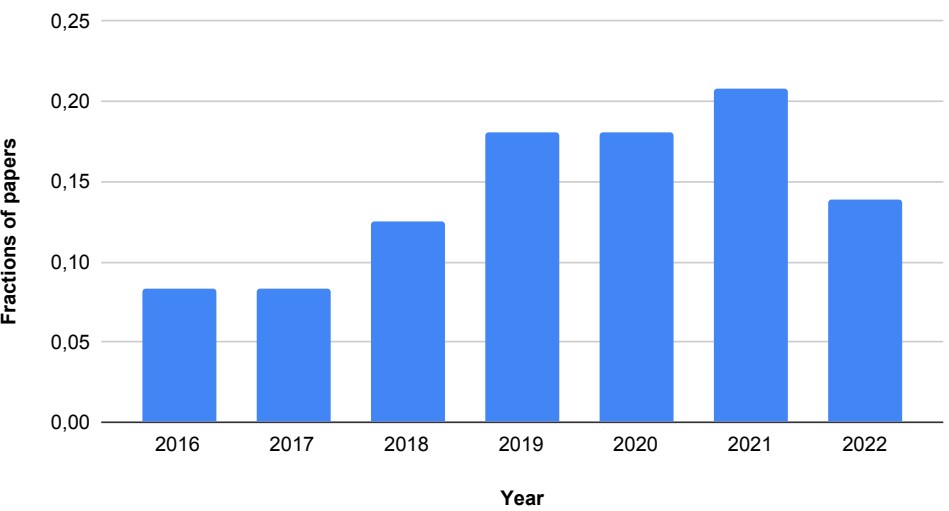

**Figure 3** Distribution of the considered papers over the last years.

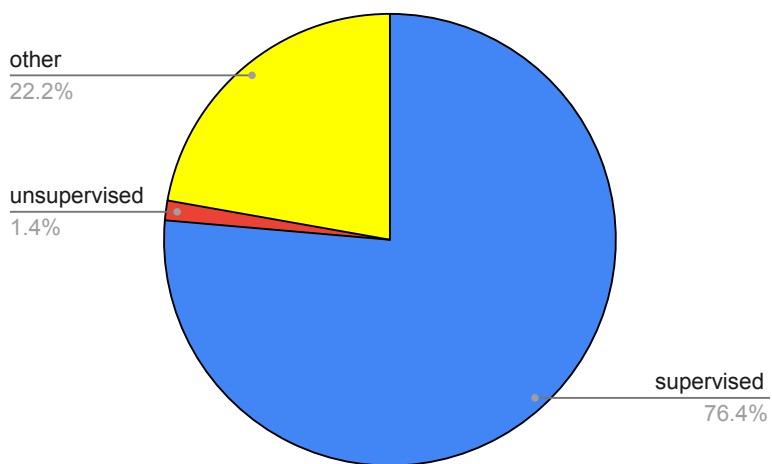

**Figure 4** Pie chart of the survived papers according to a coarse classification of the AI techniques.

### RQ1: What are the main AI techniques used for the classification and identification of the most relevant thyroid diseases?

This research question regarded the identification of the various AI techniques, as well as the main tackled thyroid diseases, used in the 72 papers identified after the filtering process. Figure 4 summarizes the percentage of the identified papers as regards a raw classification of the AI techniques. This classification entails three main classes, namely, supervised, unsupervised, and other methods. The first ones regard techniques wherein the classification is performed by exploiting labels that are already known, the second ones try to find out a classification with no prior knowledge of any label, the third ones take

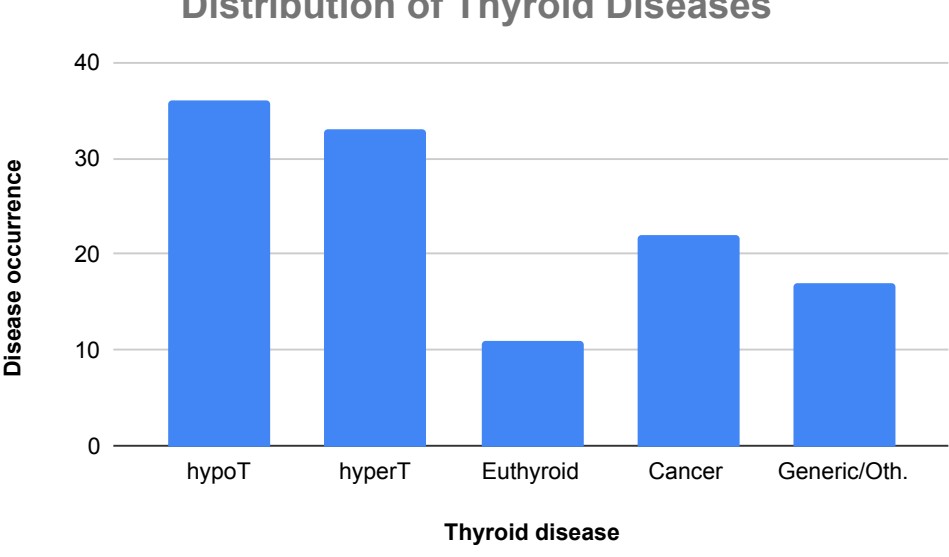

## Distribution of Thyroid Diseases

**Figure 5** Absolute occurrences of the tackled thyroid diseases in the considered papers.

advantage of either a mix of both of the aforementioned or approaches which cannot be framed in the previous categories.

From the figure, it can be noted that the most of the papers (76.39%) entails supervised techniques, while unsupervised approaches are a very small fraction (1.41%). Conversely, other or hybrid methodologies are quite relevant, reaching a percentage equal to 22.22%.

In Fig. 5, there is a classification of the considered AI techniques on the basis of the four main thyroid disease categories we have considered. In this case, a study may face more than one disease, thus the number of occurrences is not computed over 72 papers, but over 119 tackled diseases (36 papers face cases of hypothyroidism, 33 papers face cases of hyperthyroidism, 11 papers regard euthyroid disease, 22 papers face cases of thyroid cancer, and 17 papers discuss generic thyroid diseases or other thyroid-related illnesses).

In Table 2, we distribute the surveyed papers according to both the AI techniques and the just mentioned tackled thyroid diseases. The table also groups the various AI techniques according to the raw taxonomy depicted in Fig. 1.

In Fig. 6, we show a classification of the considered AI taxonomy. Also in this case, a study may exploit more than one technique, thus the relative fraction is not computed over 72 papers, but over 140 used techniques, considering repetitions in each AI group (*i.e.,* CNNs are used in 17 papers, SVM is used in 17 papers, KNN in 10 papers). As one can see, the most used techniques to face thyroid diseases are neural networks, building up more than one third of the overall AI-based methods (35%), followed by Decision Trees (16.43%), kernel-based methods (12.86%), probabilistic techniques (11.43%), ensemble

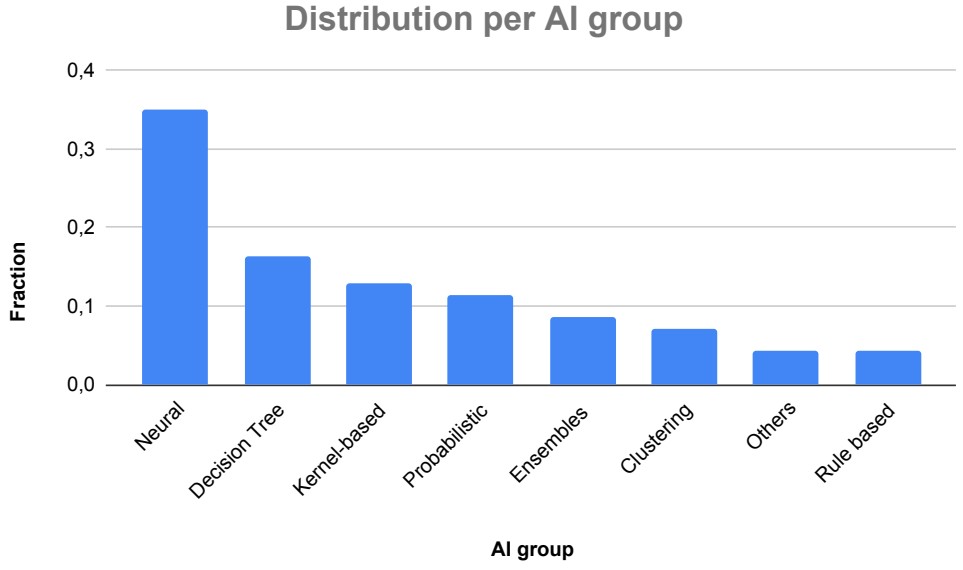

**Figure 6** Fractions of the various AI groups.

techniques (8.57%), and cluster-based methods (7.14%). The other methods represent a fraction smaller than 7%.

Finally, Table 3 presents the average, standard deviation, maximum, and minimum values of the main metrics for classification (accuracy, precision, recall, and F1-score) obtained by the considered AI techniques in the surveyed papers. As one can see, average accuracy is rather high (96.84%) with a small standard deviation, whereas F1-score is high (90.39%), but with a great standard deviation (about 9%).

In the following subsections, we briefly summarize each surveyed paper according to the AI group presented in Table 2. In case more than one AI group is used in a paper, this is described in the relative first section of appearance.

### Probabilistic approaches

In *Rao & Renuka (2020)*, Naive Bayes and a Decision Tree, built using the ID3 algorithm, are used to perform a binary prediction about whether the patient is affected by hypo- or hyperthyroidism. The authors of the study in *Pasha & Mohamed (2020)* perform feature selection on the UCI dataset for thyroid disease detection by exploiting both a Random Forest-based method and a Gain Ratio technique. In the end, the prediction is performed by comparing also different machine learning techniques, namely K-Nearest-Neighbor, Logistic Regression, and Naive Bayes. Similarly, *Houari et al. (2016)* try to reduce redundant dimensions by exploiting Copulas and LU-decomposition techniques. They test their methods also on the UCI thyroid dataset for detecting both hypo- and hyperthyroidism. For the evaluation of the data reduction techniques, the authors employed Naive Bayes, besides Artificial Neural Networks (ANN) and k-nearest neighbors (k-NN) as learning algorithms. The contribution in *Chandel et al. (2016)* compares different ML techniques in detecting thyroid-related diseases (*i.e.,* hypo- and hyperthyroidism), with a particular

Aversano et al. (2023), *PeerJ Comput. Sci.*, DOI 10.7717/peerj-cs.1394

**Table 2 Cross classification of the considered papers: according to the tackled thyroid disease and according to the various AI techniques, grouped also per macro-category.**

| | | Thyroid diseases | | | | |
|---|---|---|---|---|---|---|
| **AI group** | **AI technique** | **Hypothyroidism** | **Hyperthyroidism** | **Euthyroid disease** | **Thyroid cancer** | **Generic/others** |
| | **Naive BayesNB** | *Rao & Renuka (2020), Pasha & Mohamed (2020), Houari et al. (2016), Chandel et al. (2016), Duggal & Shukla (2020), Peya, Chumki & Zaman (2021), Riajuliislam, Rahim & Mahmud (2021), Juneja (2022)* | *Rao & Renuka (2020), Pasha & Mohamed (2020), Houari et al. (2016), Chandel et al. (2016), Duggal & Shukla (2020), Peya, Chumki & Zaman (2021), Juneja (2022)* | *Duggal & Shukla (2020), Islam et al. (2022)* | *Qin et al. (2021)* | *Juneja (2022), Kishor & Chakraborty (2021)* |
| | **Log regression LR** | *Pasha & Mohamed (2020) Riajuliislam, Rahim & Mahmud (2021)* | *Pasha & Mohamed (2020)* | | *Qin et al. (2021)* | *Raisinghani et al. (2019), Chaubey et al. (2021)* |
| **Probabilistic approaches** | **Time Series Regression TSR** | *Chandio et al. (2016)* | *Chandio et al. (2016)* | *Chandio et al. (2016)* | | |
| | **Support Vector Machine SVM** | *Duggal & Shukla (2020), Shahid et al. (2019), Ahmed & Soomrani (2016), Tyagi, Mehra & Saxena (2018), Shen et al. (2016) Li et al. (2019b), Chandel et al. (2016), Zarin Mousavi, Mohammadi Zanjireh & Oghbaie (2020), Pavya & Srinivasan (2017), Riajuliislam, Rahim & Mahmud (2021)* | *Duggal & Shukla (2020), Shahid et al. (2019), Ahmed & Soomrani (2016), Tyagi, Mehra & Saxena (2018), Shen et al. (2016), Li et al. (2019b), Chandel et al. (2016), Pavya & Srinivasan (2017)* | *Ahmed & Soomrani (2016), Shen et al. (2016), Li et al. (2019b), Duggal & Shukla (2020), Islam et al. (2022)* | *Raghavendra et al. (2018), Prochazka et al. (2019), Raghavendra et al. (2017), Qin et al. (2021), Shen et al. (2021)* | *Tyagi, Mehra & Saxena (2018), Kaur, Kumar & Kumar (2019) Kishor & Chakraborty (2021)* |
| **Kernel-based approaches** | **multi-kernel SVM** | *Kumar (2020)* | *Kumar (2020)* | *Kumar (2020)* | | |
| | **Decision Tree DT** | *Rao & Renuka (2020), Sidiq & Mutahar Aaqib (2019), Tyagi, Mehra & Saxena (2018), Hayashi (2017), Peya, Chumki & Zaman (2021), Riajuliislam, Rahim & Mahmud (2021), Juneja (2022)* | *Rao & Renuka (2020), Sidiq & Mutahar Aaqib (2019), Tyagi, Mehra & Saxena (2018), Hayashi (2017), Peya, Chumki & Zaman (2021), Juneja (2022)* | *Islam et al. (2022)* | *Hao et al. (2018)* | *Tyagi, Mehra & Saxena (2018), Raisinghani et al. (2019), Kaur, Kumar & Kumar (2019), Chaubey et al. (2021), Juneja (2022), Kishor & Chakraborty (2021)* |
| | **Decision stump DS** | | *Jha et al. (2018)* | | | |
| | **Random Forest RF** | *Sidiq & Mutahar Aaqib (2019), Shahid et al. (2019), Duggal & Shukla (2020), Riajuliislam, Rahim & Mahmud (2021), Juneja (2022)* | *Sidiq & Mutahar Aaqib (2019), Shahid et al. (2019), Imbus et al. (2017), Duggal & Shukla (2020), Juneja (2022)* | *Duggal & Shukla (2020) Islam et al. (2022)* | *Prochazka et al. (2019) Qin et al. (2021)* | *Pan et al. (2016), Prochazka et al. (2019), Raisinghani et al. (2019), Kaur, Kumar & Kumar (2019), Juneja (2022)* |

**Table 2** (*continued*)

| | | Thyroid diseases | | | | |
|---|---|---|---|---|---|---|
| AI group | AI technique | Hypothyroidism | Hyperthyroidism | Euthyroid disease | Thyroid cancer | Generic/others |
| Decision Tree approaches | Iterative Dichotomiser 3ID3 | *Zarin Mousavi, Mohammadi Zanjireh & Oghbaie (2020)* | | | | |
| | Chi-squared Automatic Interaction Detection-CHAID | *Zarin Mousavi, Mohammadi Zanjireh & Oghbaie (2020)* | | | | |
| | Recursive-Rule extraction Re-RX | *Hayashi (2017)* | *Hayashi (2017)* | | | |
| | Case-based reasoning | *Bentaiba-Lagrid et al. (2020)* | *Bentaiba-Lagrid et al. (2020)* | | | |
| | Rutherford Backscattering Spectrometry RBS | | *Imbus et al. (2017)* | | | |
| Rule-based approaches | Fuzzy RBS | *Asaad Sajadi et al. (2019), Kumari & Sharma (2019)* | *Kumari & Sharma (2019)* | | | |
| | Artificial neural network ANN | *Sidiq & Mutahar Aaqib (2019), Mahurkar & Gaikwad (2017), Tyagi, Mehra & Saxena (2018), Houari et al. (2016), Vivar et al. (2020) Zarin Mousavi, Mohammadi Zanjireh & Oghbaie (2020)* | *Sidiq & Mutahar Aaqib (2019), Mahurkar & Gaikwad (2017), Tyagi, Mehra & Saxena (2018), Houari et al. (2016), Vivar et al. (2020)* | *Islam et al. (2022)* | *Ahmed et al. (2022), Cordes et al. (2021), Jin et al. (2021)* | *Tyagi, Mehra & Saxena (2018), Raisinghani et al. (2019), Kaur, Kumar & Kumar (2019), Santos et al. (2019), Kishor & Chakraborty (2021)* |
| | Bidirectional Long Short-Term Memory - Long Short Term Memory (LSTM) BLSTM-LSTM | *Yue et al. (2020)* | *Yue et al. (2020), Lu et al. (2020)* | | | *Chai (2020)* |
| | Convolutional Neural Network CNN | *Yue et al. (2020) Ananthi et al. (2022) Khan (2021)* | *Yue et al. (2020) Ananthi et al. (2022)* | | *Yin et al. (2019), Yi et al. (2017), Lyu & Haque (2018), Li et al. (2019a), Moran et al. (2018), Ananthi et al. (2022), Chu, Zheng & Zhou (2021) Liu et al. (2021), Santillan et al. (2021), Song et al. (2022)* | *Poudel et al. (2019), Guo & Du (2019) Ananthi et al. (2022) Pi et al. (2022), Yang et al. (2021)* |

Aversano et al. (2023), *PeerJ Comput. Sci.*, DOI 10.7717/peerj-cs.1394

**Table 2** (*continued*)

| | | Thyroid diseases | | | | |
|---|---|---|---|---|---|---|
| AI group | AI technique | Hypothyroidism | Hyperthyroidism | Euthyroid disease | Thyroid cancer | Generic/others |
| | **Generative Adversarial Network GAN** | *Zhang, Huang & Lv (2020)* | *Zhang, Huang & Lv (2020)* | | *Shi et al. (2020), Zhao et al. (2022)* | |
| | **Capsule Network CN** | | | | *Ai et al. (2022)* | |
| | **RBF NN** | *Juneja (2022)* | *Juneja (2022)* | | | *Juneja (2022)* |
| | **Autoencoder** | *Saktheeswari & Balasubramanian (2021)* | *Saktheeswari & Balasubramanian (2021)* | | *Saktheeswari & Balasubramanian (2021)* | |
| | **Recurrent Neural Network RNN** | | | | *Santillan et al. (2021)* | |
| **Neural Network** | **Multilayer Perceptron MLP Back Propagation** | *Yue et al. (2020), Zarin Mousavi, Mohammadi Zanjireh & Oghbaie (2020), Pavya & Srinivasan (2017), Hosseinzadeh et al. (2021), Juneja (2022)* | *Yue et al. (2020), Pavya & Srinivasan (2017), Hosseinzadeh et al. (2021), Juneja (2022)* | *Qin et al. (2021)* | *Juneja (2022)* | |
| **Cluster-based approaches** | **K-Nearest Neighbors Algorithm KNN** | *Shahid et al. (2019), Tyagi, Mehra & Saxena (2018), Pasha & Mohamed (2020), Houari et al. (2016), Chandel et al. (2016), Pasha & Mohamed (2020), Peya, Chumki & Zaman (2021)* | *Shahid et al. (2019), Tyagi, Mehra & Saxena (2018), Pasha & Mohamed (2020), Houari et al. (2016), Chandel et al. (2016), Peya, Chumki & Zaman (2021)* | *Islam et al. (2022)* | *Qin et al. (2021)* | *Tyagi, Mehra & Saxena (2018), Kaur, Kumar & Kumar (2019) Chaubey et al. (2021) Kishor & Chakraborty (2021)* |

Peerj Computer Science

**Table 2** (*continued*)

| | | Thyroid diseases | | | | |
|---|---|---|---|---|---|---|
| **AI group** | **AI technique** | **Hypothyroidism** | **Hyperthyroidism** | **Euthyroid disease** | **Thyroid cancer** | **Generic/others** |
| | **AdaBoost** | *Zarin Mousavi, Mohammadi Zanjireh & Oghbaie (2020), Yadav & Pal (2022), Priyadharsini & Sasikala (2022)* | *Priyadharsini & Sasikala (2022)* | *Islam et al. (2022)* | | *Kishor & Chakraborty (2021)* |
| | **Stacking** | *Yadav & Pal (2022) Sidiq & Mutahar Aaqib (2019)* | *Sidiq & Mutahar Aaqib (2019), Jha et al. (2018)* | | | |
| | **Bagging** | *Zarin Mousavi, Mohammadi Zanjireh & Oghbaie (2020), Yadav & Pal (2022) Priyadharsini & Sasikala (2022)* | *Priyadharsini & Sasikala (2022)* | | *Qin et al. (2021)* | |
| | **Vote ensemble** | *Sidiq & Mutahar Aaqib (2019), Yadav & Pal (2022)* | *Sidiq & Mutahar Aaqib (2019), Jha et al. (2018)* | *Islam et al. (2022)* | *Yin et al. (2019)* | *Pan et al. (2016)* |
| **Ensembles** | **Dynamic ensemble** | *Alam, Siddique & Adeli (2020)* | *Alam, Siddique & Adeli (2020)* | *Alam, Siddique & Adeli (2020)* | | |
| | **Semi supervised Learning** | | | | | |
| | **Weighted Extreme Learning Machine WELM** | | | | | *Priya & Manavalan (2018)* |
| | **Extreme Learning Machine ELM** | *Pavya & Srinivasan (2017), Ma et al. (2018) Juneja (2022)* | *Pavya & Srinivasan (2017), Ma et al. (2018) Juneja (2022)* | | | *Juneja (2022)* |
| **Other Learning approaches** | **Amended fused TOPSIS-VIKOR for classification ATOVIC** | *Baccour (2018)* | *Baccour (2018)* | | | |

**Table 3  Statistics of the metrics of the surveyed AI techniques for thyroid disease classification.**

|  | Accuracy (%) | Precision (%) | Recall (%) | F1-score (%) |
|---|---|---|---|---|
| Average | 96.84 ± 1.91 | 94.96 ± 6.32 | 90.21 ± 11.90 | 90.39 ± 9.02 |
| Maximum | 100.00 | 100.00 | 100.00 | 99.00 |
| Minimum | 93.00 | 85.00 | 69.00 | 74.00 |

focus on Naive Bayes, K-Nearest Neighbor, and Support Vector Machine. *Juneja (2022)* presents a fuzzy adaptive feature filtration and expansion-based model to generate a novel feature set related to thyroid. The obtained feature set is then analyzed through Extreme Learning Machine classifiers, whose performance is compared with Naïve Bayes, Decision Tree, Multilayer Perceptron, and Radial Basis Function networks. *Kishor & Chakraborty (2021)* compare seven machine learning classifiers such as decision tree, support vector machine, Naïve Bayes, adaptive boosting, Random Forest, artificial neural networks, and K-nearest neighbor. They find out that Random Forest is the best performing algorithm to predict fatal diseases about thyroid. Authors in *Qin et al. (2021)* try to study papillary thyroid cancer through the analysis of magnetic resonance imaging radiomics by exploiting eight classifiers (logistic regression, bagging, Random Forests, extremely randomized trees, support vector machines, Naïve Bayes, multilayer perception, and K-nearest neighbors). Some of the models succeeded into reaching a performance of correct classification higher than 95%.

In *Peya, Chumki & Zaman (2021)*, a thyroid disease prediction model is proposed exploiting three machine learning classification algorithms (*i.e.,* K-Nearest Neighbor, Naive Bayes, and Decision Trees). Using the thyroid data of the UCI machine learning repository and a 10-fold cross-validation, the decision tree resulted the most accurate algorithm, with a 99.7% of accuracy.

*Riajuliislam, Rahim & Mahmud (2021)* try to predict early stage hypothyroidism. They apply three different feature selection procedures: Principal Component Analysis (PCA), Recursive Feature Elimination (RFE), and Univariate Features Selection (UFS). Moreover, they use different classification algorithms (*i.e.,* support vector machine, decision tree, random forest, logistic regression, and Naive Bayes). RFE applied to the UCI thyroid dataset allowed the authors to achieve a constant 99% accuracy value over the various considered classification algorithms.

In *Raisinghani et al. (2019)*, the authors compare different machine learning approaches, (logistic regression, decision trees, random forest, support Vector Machine) to develop predictive models to detect a generic thyroid disease.

The contribution in *Chandio et al. (2016)* exploits time series regression to create an intelligent system for thyroid disease visualization. This allows a careful surveillance of the thyroid disease, with a particular focus on hypothyroidism, hyperthyroidism, and euthyroid disease.

The study in *Chaubey et al. (2021)* compares logistic regression, decision tree, and kNN on the UCI knowledge discovery database for thyroid diseases detection. The best result of accuracy is obtained by applying the kNN classifier (96.87%).

Finally, *Islam et al. (2022)* try to detect euthyroid disease by leveraging eleven machine learning classifiers, namely, Gaussian naive Bayes, ANN, CatBoost, XGBoost, Random Forest, LightGBM, Decision Tree, Extra-trees, SVC, and KNN, and applying also feature selection to the sick-euthyroid dataset. The best performing classifier resulted to be ANN with a maximum accuracy equal to 95.87%.

### Kernel-based approaches

*Duggal & Shukla (2020)* perform feature selection and extraction before applying naive Bayes, Support Vector Machine, and Random Forest to identify hypothyroidism, hyperthyroidism, and euthyroid disease.

The authors of *Shahid et al. (2019)* compare Random Forest, Support Vector Machine, and K-Nearest Neighbours on the UCI thyroid dataset, to discover the best performing algorithm, resulted to be Random Forest, in detecting hypo- and hyperthyroidism.

The contribution in *Ahmed & Soomrani (2016)* provides a framework, named Thyroid Disease Types Diagnostics (TDTD), aiming at making a diagnosis of various thyroid diseases in a very structured and transparent manner and exploiting binary and multi-SVM algorithms, as well as Bayesian isotonic regression for missing values. In *Tyagi, Mehra & Saxena (2018)*, the authors present again the results in detecting hypo- and hyperthyroidism (by using the UCI dataset) through different machine learning techniques, such as decision trees, artificial neural networks, support vector machines, and k-nearest neighbors.

*Kumar (2020)* introduces a novel mutliclass SVM approach to detect four types of subjects, *i.e.,* people affected by hypothyroidism, hyperthyroidism, euthyroidism disease and healthy euthyroid, on the already mentioned UCI dataset. *Shen et al. (2016)* propose a novel scheme to optimize the parameters of SVM by means of the fly optimization algorithm. The novel scheme is compared with other optimization algorithms and its efficiency is tested on four datasets, including the UCI one about thyroid.

In *Li et al. (2019b)*, a novel optimization technique for SVM applied to the UCI thyroid dataset is proposed. It is based on the teaching-learning algorithm and differential evolution and it permitted to SVM to reach better performance in comparison with other solutions.

Malignant nodules detection is the focus of the work in *Raghavendra et al. (2018)*. The paper presents a computer-aided diagnosis system to detect thyroid malignant nodules by means of higher order spectral entropy features and using particle swarm optimization (PSO) and support vector machine (SVM) frameworks.

In *Prochazka et al. (2019)*, the authors propose a computer-aided diagnosis system using direction independent features of ultrasound images of the thyroid gland, with the aim of detecting malignant nodules by means of Random Forest and SVM.

Similarly, the authors of the study in *Raghavendra et al. (2017)* try to propose an SVM-based computer-aided diagnosis system, by exploiting fusion of Spatial Gray Level Dependence Features (SGLDF) and fractal textures to detect benign and malignant thyroid lesions in ultrasound images.

*Kaur, Kumar & Kumar (2019)* propose an IoT-based framework using SVM, Random Forest, Decision Trees, k-nearest neighbour, and artificial neural networks to be tested on different healthcare datasets, including one about general thyroid disease.

*Shen et al. (2021)* apply SVM to platelet RNA-seq data in order to differentiate different types of thyroid cancer and are able to achieve an accuracy of 97%.

### Decision Tree approaches

*Hayashi (2017)* tries to create a white-box model to make prediction on the UCI thyroid datasets by exploiting the synergical effects of Recursive-Rule eXtraction (Re-RX) with J48 graft in terms of rule definition in the IF-THEN form for predicting both hypo- and hyperthyroidism. In *Sidiq & Mutahar Aaqib (2019)*, the authors employ decision trees, random forest, vote ensemble, and stack ensemble in order to detect both hypo- and hyperthyroidism. *Hao et al. (2018)* introduce a decision tree improved by MS-Apriori for the prognosis of lymph node metastasis (LNM) in patients with thyroid cancer. MS-Apriori is used to generate association rules considering rare items by multiple supports and fuzzy logic is introduced to map attribute values to different sub-intervals. The used dataset is made of clinical-pathological data, obtained from the First Hospital of Jilin University.

*Jha et al. (2018)* present a hybrid algorithm for healthcare data mining by using Decision Stump (DS), StackingC (SC), and voting methods to tackle the hyperthyroidism issue. In *Imbus et al. (2017)*, a random tree and a rule-based classifier (JRip) are used to detect primary hyperparathyroidism; while in *Pan et al. (2016)*, random forest, together with principal component analysis and rotation transformation, is used to detect a general thyroid disease by exploiting the UCI thyroid dataset.

Finally, *Zarin Mousavi, Mohammadi Zanjireh & Oghbaie (2020)* apply computational methods based on decision trees, like ID3 and CHAID, SVM and multi-layer perceptrons, enriched with bagging and boosting techniques, for the identification of congenital hypothyroidism.

### Rule-based approaches

In *Bentaiba-Lagrid et al. (2020)*, a new amplification technique, based on randomization for systems incorporating a structured case-base that speeds up case retrieval while supporting case retention, is presented. The case-base is segmented in a novel manner, with new similarity functions based on features' weights in order to accelerate the retrieval of the case-base reasoning. The system was applied to the detection of both hypo- and hyperthyroidism and compared with different machine learning methods.

In *Asaad Sajadi et al. (2019)*, hypothyroidism is detected by means of a fuzzy rule-based expert system, which is proved to perform better than a logistic regression model on a real dataset.

Finally, the authors of *Kumari & Sharma (2019)* draft a fuzzy logic-based expert system to tackle both hypo- and hyperthyroidism.

### Neural networks

*Mahurkar & Gaikwad (2017)* exploit artificial neural networks in conjunction with K-Means to normalize raw data for hypo- and hyperthyroidism. In *Vivar et al. (2020)*, a guiding computer-aided diagnosis system, using a neural network with a dropout at the input layer, and integrated gradients of the trained network at test-time to attribute feature

importance dynamically, is proposed. The technique is applied also to the UCI thyroid dataset to detect both hypo- and hyperthyroidism.

Similarly, in *Santos et al. (2019)*, the authors introduce a decision support system, to make general thyroid dysfunction assessment, based on an artificial neural network and complemented by a novel approach to knowledge representation and argumentation. In *Yue et al. (2020)*, Fourier transform infrared spectroscopy was combined with three neural network models, namely multilayer perceptron, long-short-term memory network, and a convolutional neural network in order to detect hypo- and hyperthyroidism.

*Chai (2020)* tackles thyroid disease in general by means of knowledge graphs, by extracting the relationships between bio-medical entities for feeding a bidirectional long short-term memory network. This combination proved to have better diagnostic effects than other techniques on an image dataset from the university of Shanghai. In *Zhang, Huang & Lv (2020)*, the authors propose a synthetic data augmentation method based on progressive generative adversarial network in order to improve the performance in deep learning detection of hypo- and hyperthyroidism.

*Yin et al. (2019)* propose a hybrid cutting network, featuring a regional attribute cutting method, for feature extraction and classification applied to a dataset of thyroid ultrasound images. The objective was to detect malignant thyroid nodules that could cause cancer.

In *Yi et al. (2017)*, the authors propose a novel diagnostic system to detect thyroid cancer, with a particular focus on thyroid nodule risk assessment. The method employs convolutional neural networks to analyze ultrasound images.

The contribution in *Lyu & Haque (2018)* embeds high dimensional RNA-Sequence data into bi-dimensional images and uses a convolutional neural network to detect various types of cancer, including thyroid cancer.

In *Li et al. (2019a)*, the authors employ deep convolutional neural networks to enhance the diagnostic accuracy of thyroid cancer through the analysis of sonographic images coming from clinical ultrasounds. Ultrasound images are also used in *Poudel et al. (2019)* to feed convolutional neural networks for texture classification of anatomical structures of the thyroid for detecting general changes of its shape. Conversely, in *Moran et al. (2018)*, thermograms are exploited by convolutional neural networks for the early identification of thyroid nodules that can possibly cause cancers. *Guo & Du (2019)* exploit again ultrasound images of the standard plane of the thyroid to evaluate its general status by means of deep convolutional neural networks. In particular, a 18-layer ResNet achieves the best results according to the authors. In *Shi et al. (2020)*, the authors integrate domain knowledge, extracted from standardized terminology, and deep learning (Auxiliary Classifier Generative Adversarial Network) into a synthetic medical image augmentation technique to classify ultrasonography thyroid nodules.

In *Lu et al. (2020)*, hyperthyroidism is tackled, with a particular focus on its progression, by means of enhanced LSTMs with an adaptive loss function. The analyzed data regard blood test information in the early stage from a Shangai hospital.

*Cordes et al. (2021)* detect thyroid malignant nodules using artificial neural networks on ultrasonographic characteristics obtaining an accuracy of 84.4%. Similarly, *Jin et al. (2021)* analyze clinical ultrasound imaging data, in five hospitals in China, through artificial neural

networks obtaining performance between 80% and 90%. *Ahmed et al. (2022)* use artificial deep neural networks on the concatenation of 6 databases containing data collected from the Garvan Institute in Sydney, Australia, to discover thyroid cancer. The main and only shown result is the accuracy of 98%; other results and the optimization process are not described in detail.

*Ananthi et al. (2022)* and *Khan (2021)* exploit convolutional neural networks to try to detect the onset of thyroid dysfunctions. The former employs X-ray images to prevent hypo- and hyperthyroidism as well as malignant nodules and other dysfunctions reaching a 99% of accurate prediction. The latter only focus on hypothyroidism and achieve 98% of accuracy. *Liu et al. (2021)* tackle the issue of malignant nodes identification using convolutional neural networks as well. In this case, they apply information fusion techniques on ultrasound images and radio frequency signals, achieving better results than using only ultrasound images in detecting malignant thyroid nodules.

In *Santillan et al. (2021)* and *Song et al. (2022)* the focus is on the detection of malignant thyroid nodules which could possibly generate cancer. The former apply convolutional and recurrent neural networks to Fourier Transform infrared spectroscopy data achieving accuracy of 98.06%. The latter employ a feature-enhanced dual branch convolutional neural network on ultrasound images of the thyroid gland, obtaining the best mean average precision of identification equal to 92.5%. *Chu, Zheng & Zhou (2021)* face the issue of malignant thyroid nodule detection too, by employing a mark-guided ultrasound deep network segmentation model which, in turn, is based on different types of convolutional neural networks. They achieve a segmentation accuracy equal to 97.85%, improving the outcomes of other standard convolutional neural networks.

*Yang et al. (2021)* use convolutional neural networks on thyroid scintigrams to detect general dysfunctions of the gland and report an accuracy of 92.73%. Similarly, *Pi et al. (2022)* exploit convolutional neural networks and the fusion of deep and handcrafted features from thyroid scintigraphy to early detect thyroid dysfunctions. They were able to reach an accuracy equal to 91.18% and an f-measure equal to 88.11%.

In *Zhao et al. (2022)*, semantic consistency generative adversarial network are employed to detect malignant thyroid nodules using ultrasound data. The proposed method is claimed to reach an accuracy equal to 94.30% and an area under the curve equal to 97.02%.

*Hosseinzadeh et al. (2021)* use a multiple multilayer perceptron neural network to identify hypo- and hyperthyroidism in the context of Internet of Medical Things, reaching an accuracy of 99%. *Saktheeswari & Balasubramanian (2021)* exploit an autoencoder-based neural network using also a multi-layer tree-based state machine to detect malignant thyroid nodules as well as hypo- and hyperthyroidism, with a final mean accuracy equal to 98.90%.

Finally, *Ai et al. (2022)* apply a recent type of neural networks, *i.e.,* capsule networks, on ultrasonic thyroid images to detect possible thyroid cancer traces achieving a top accuracy equal to 81.06%.

### Ensembles

In *Yadav & Pal (2022)*, an ensemble of different machine learning techniques is employed to detect thyroid hormone disease. The ensemble uses Boosting, Bagging, Stacking, and Voting and it is aimed at identifying hypothyroidism patients.

In *Alam, Siddique & Adeli (2020)*, the authors present a novel dynamic ensemble learning of neural networks. It provides an automatic design of the ensemble, a maintaining of accuracy and diversity of the composing neural networks, and very few parameters to be designed by the user. The technique is successfully employed to detect hypothyroidism, hyperthyroidism, and euthyroid disease.

*Priyadharsini & Sasikala (2022)* exploit Adaboost and Bagging as ensemble machine learning methods to correctly detect hypo- and hyperthyroidism. Bagging resulted in better performance as regards all the main metrics: accuracy 99.20%, precision 99.9%, f-measure 99.8%.

Finally, *Yadav & Pal (2022)* exploit Boosting, Bagging, Stacking, and Voting ensembles to detect hypothyroidism achieving a top accuracy equal to 99.86% and a top recall equal to 99.88%.

### Other learning approaches

In *Priya & Manavalan (2018)*, a weighted extreme learning machine technique hybridized with Invasive Weed optimization is used to detect general thyroid disease. Conversely, in *Pavya & Srinivasan (2017)*, filter-based and wrapper-based feature selection methods are applied to four classifiers, namely, MultiLayer Perceptron Back Propagation Neural Network, Support Vector Machine, and Extreme Learning Machine, in order to detect both hypo- and hyperthyroidism. In *Ma et al. (2018)*, the authors introduce a novel hybrid diagnosis system, integrating local fisher discriminant analysis and kernelized extreme learning machine method for thyroid disease diagnosis (hypo- and hyperthyroidism). Finally, in *Baccour (2018)*, a new classification system for both hypo- and hyperthyroidism and based on fused VIKOR and TOPSIS is proposed.

## RQ2: What datasets about thyroid diseases are used in the considered AI solutions?

In this section, we describe the main characteristics of the datasets used in the surveyed papers.

In Fig. 7, we show the availability of the datasets used in the papers we considered in this systematic review. As one can see, the number of private or non-available datasets is higher (55.42%) compared with the number of publicly available ones.

Looking at the public datasets, we observed that the most used dataset is the UCI one (https://archive.ics.uci.edu/ml/datasets/Thyroid+Disease), exploited 27 times (*Duggal & Shukla, 2020*; *Shahid et al., 2019*; *Pan et al., 2016*; *Pavya & Srinivasan, 2017*; *Mahurkar & Gaikwad, 2017*; *Ahmed & Soomrani, 2016*; *Tyagi, Mehra & Saxena, 2018*; *Kumar, 2020*; *Pasha & Mohamed, 2020*; *Shen et al., 2016*; *Bentaiba-Lagrid et al., 2020*; *Raisinghani et al., 2019*; *Vivar et al., 2020*; *Li et al., 2019b*; *Ma et al., 2018*; *Kour, Manhas & Sharma, 2020*; *Khan, 2021*; *Priyadharsini & Sasikala, 2022*; *Peya, Chumki & Zaman, 2021*; *Chaubey et al., 2021*; *Hosseinzadeh et al., 2021*; *Juneja, 2022*; *Kishor & Chakraborty, 2021*; *Islam et al., 2022*;

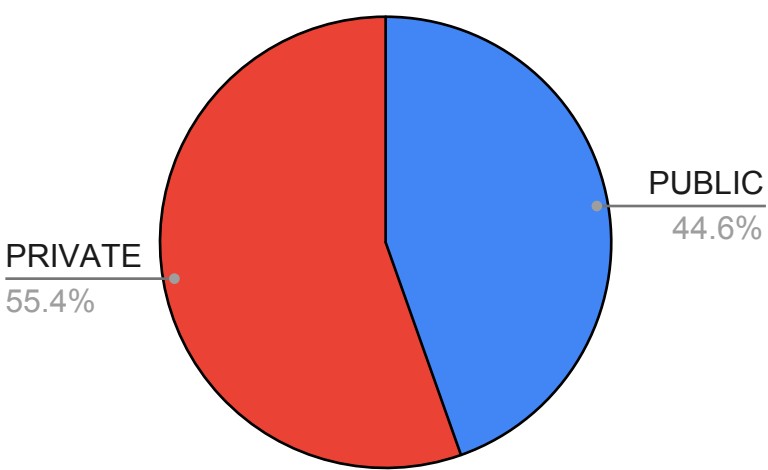

**Figure 7** Availability of the datasets in the surveyed papers.

*Saktheeswari & Balasubramanian, 2021*; *Chandel et al., 2016*; *Priya & Manavalan, 2018*). The UCI dataset is characterized by 7,200 instances and 21 categorical and real attributes. It is quite old since it was built in 1987.

The study proposed in *Rao & Renuka (2020)* uses a dataset available on the Kaggle machine learning website (https://www.kaggle.com/datasets/kumar012/hypothyroid). This dataset consists of the 3,162 instances, including basic patient information details and clinical history. The considered 27 attribute values are boolean or continuous.

A public dataset (http://visual.ic.uff.br/en/thyroid/) is also used by *Moran et al. (2018)*. This dataset contains the infrared image, clinical, and histopathological data collected from patients with thyroid nodules at the Outpatient Clinic of Endocrinology and Surgery of the Hospital Universitario Antonio Pedro—HUAP of the Fluminense Federal University (UFF). The number of instances considered in the study is 92 with 20 × 20 featuring images.

Finally, in Table 4, we present some statistics about the main characteristics of the datasets used in the surveyed papers. They show that the average number of instances per dataset is about 5,641 samples, while the average number of features is almost 26,000, but in this case we have to consider that for image datasets the number of features was considered equal to the number of pixels, thus making the statistics increase very much.

### RQ3: What data types are used to detect and classify thyroid diseases using the considered AI techniques?

In this section, we analyze the types of features considered in the surveyed papers.

In Fig. 8, we show a pie chart to highlight the different percentages of data used in the surveyed papers. As it can be easily inferred, the majority of the considered features (51.85%) is made of clinical data, *i.e.,* data mainly coming from the blood analysis, while

**Table 4  Statistics of the number of instances and of features in the considered datasets.**

|  | Instances | Features |
|---|---|---|
| Average | 5641 | 25,674 |
| Maximum | 92,062 | 711,680 |
| Minimum | 92 | 4 |

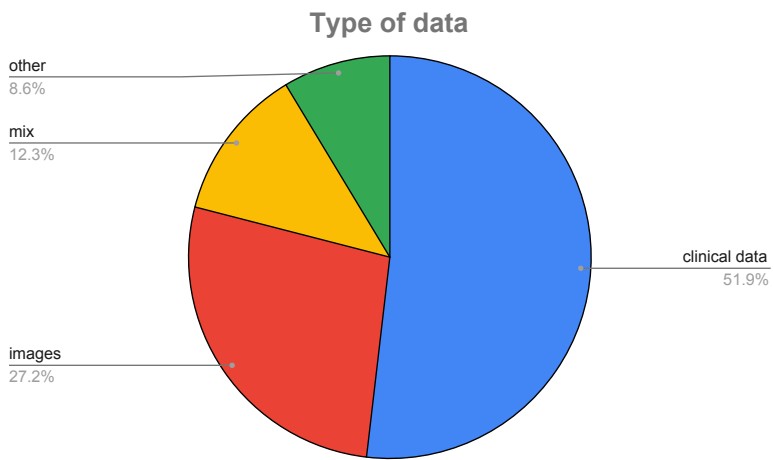

**Figure 8  Type of data used in the considered datasets.**

27.16% of the surveyed papers use images of the thyroid, and only a small percentage (12.35%) exploit both types of data.

Table 5 shows the occurrences of the ten most used features in the surveyed papers. As it is evident, the most used feature is the thyroid-stimulating hormone (TSH) with 39 occurrences, followed by triiodothyronine (T3) hormone, the age, and the gender of the subject. Other very used features regard the pixel value (20), the only feature used whenever image datasets are employed, as well as the thyroxine, also called T4, FTI (Free T4 Index) and T4U (T4 Uptake). Pregnancy status is the least important feature in the top ten context, and can be obviously used only in female patients.

## DISCUSSION

In this section, we perform a brief discussion about the main outcomes of the systematic review and some important open research directions.

One of the first results is that a great majority of papers focuses on supervised methods. Therefore, the development of novel and effective techniques employing also unsupervised or semi-supervised approaches is surely a need in the application of artificial intelligence techniques to the automated detection of thyroid dysfunctions and diseases. Moreover, they ensure a possible solution when there is a limited labeling budget (*Yoon et al., 2022*). In some cases, effectively, labels are particularly tedious to obtain, and labeled samples are not available during the training step. Finally, even when labeled data are available, there could be biases in the way samples are labeled, causing distribution differences.

**Table 5  Occurrences of the top ten features used in the surveyed papers.**

| Feature | Occurrences |
| --- | --- |
| TSH (thyroid-stimulating hormone) | 39 |
| T3 (triiodothyronine) | 28 |
| Age | 28 |
| Gender | 27 |
| Pixel value | 19 |
| TT4 (total thyroxine) | 19 |
| FTI (Free T4 Index) | 18 |
| T4U (T4 Uptake) | 18 |
| Pregnancy | 14 |
| T4 (thyroxine) | 14 |

We also observed that the greatest number (54 papers) of the surveyed studies is focused on offering a solution to the issue of disease detection, while a very poor number of papers faces the topic of disease prediction and monitoring. The main reason, according to our opinion, is the absence of useful datasets for this scope.

As regards the tackled diseases, there is a clear prevalence of hypo- and hyperthyroidism. Therefore, more focused scientific work, especially regarding euthyroid pathology and cancers, should be carried out, in order to foster the use of AI techniques in the detection of thyroid dysfunction.

The average performance of the surveyed AI techniques in detecting thyroid-related diseases can be further improved (Table 2 shows an average F1-score of 90.39%). Given the higher use of neural networks among the surveyed techniques, a greater focus on deep neural networks and the careful optimization of their parameters should be the topic of future research papers.

Moreover, there are few datasets publicly available, and among these, the most used one is the UCI dataset, which is quite old (1987). The need for novel, public, and updated datasets is, thus, compelling, together with the application of proper feature selection and feature reduction techniques, which are very poorly investigated in the surveyed papers.

However, based on the declarations of the authors of the analyzed studies, we observed that their main limitation is in the quality and dimensions of the adopted datasets. Some authors highlight the necessity to increase the adopted dataset (*Poudel et al., 2019*; *Yin et al., 2019*; *Moran et al., 2018*; *Kaur, Kumar & Kumar, 2019*; *Ma et al., 2018*; *Qin et al., 2021*; *Shen et al., 2021*). In other studies, authors identify as a limit the reduced heterogeneity and balancing of the adopted dataset (*Pan et al., 2016*; *Yi et al., 2017*; *Pi et al., 2022*; *Vivar et al., 2020*). Moreover, in the surveyed datasets, the usage of clinical data is prevalent; thus, also in view of the greater usage of various deep learning techniques, greater development of public image datasets to be used with proper deep convolutional neural networks should be encouraged.

Finally, even if in all considered studies there is the adoption of real data (the data are collected by medical institutions from patients), the proposed approaches are almost never

applied in a real context. This highlights that there is still a lot of resistance from medical institutions to use AI solutions to support their medical activities.

## CONCLUSIONS

In this article, we have performed a detailed systematic review about the application of artificial intelligence techniques to the study, analysis, and detection of various thyroid-related diseases and dysfunctions.

First of all, we have summarized some similar reviews and pointed out the necessity of the systematic review we have carried out.

Subsequently, we have detailed the research process we employed, describing the research questions, the used databases, and the employed queries. We have also described in detail the different filtering phases that led us to find out the 72 analyzed papers.

In the end, we have presented and discussed in depth the results of the systematic review: the main AI techniques used for the classification and identification of the most relevant thyroid diseases (RQ1), the datasets, and data types used in the considered AI solutions (RQ2, RQ3).

The obtained results show that supervised methods are very widespread (they are used in 76.39% of the analyzed papers), while unsupervised approaches are a very small fraction (1.39%).

This, in our opinion, is due to the greater interest of the medical staff in classification tasks and the presence of large labeled datasets. However, the interest about unsupervised and hybrid approaches should be also encouraged because of their capability of learning the data and classifying them without any labels.

Moreover, the most used AI technique to face thyroid diseases is neural networks, building up more than one-third of the overall AI-based methods (35%). The great interest about neural networks is motivated by the advantages of their application in medical domain. However, their ability to learn and model complex relationships is really important in the considered domain where many of the relationships between inputs and outputs are non-linear and complex.

Finally, a large number of papers study hypothyroidism and hyperthyroidism (30.25% and 27.73%). The reason for this is that hypothyroidism and hyperthyroidism are the two most common thyroid pathologies according to the most recent studies (*Muñoz-Ortiz et al., 2020*).

Looking at the adopted datasets, we observed that there is a high number of not available datasets (55.42%) and this limits the possibility to extend the existing research. However, the most used dataset (27 times in the considered papers) is the UCI one, which is quite old (it was published in 1987). Finally, we observed that the majority of the considered features (51.85%) are made of clinical data, while 27.16% of the surveyed papers use images of the thyroid, and only a small percentage (12.35%) exploit both types of data. This, according to our opinion, is due to the reduced availability of datasets containing both images and clinical data.

In the final discussion, we have pointed out the main open research directions, as well as the issues still to be solved to make artificial intelligence a viable solution for the quick diagnosis and classification of any thyroid-related disease or dysfunction.

### Funding

The work of Riccardo Pecori has been supported by the PON R&I 2014-2020 "AIM: Attraction and International Mobility" project. The work of Mario Luca Bernardi has been supported by the Italian Ministry of Education and Research (MIUR), in the framework of the "Departments of Excellence" project. The funders had no role in study design, data collection and analysis, decision to publish, or preparation of the manuscript.

### Grant Disclosures

The following grant information was disclosed by the authors:
PON R&I 2014-2020 "AIM: Attraction and International Mobility" project.
The Italian Ministry of Education and Research (MIUR), in the framework of the "Departments of Excellence" project.

### Competing Interests

Mario Luca Bernardi and Marta Cimitile are Academic Editors for PeerJ.

### Author Contributions

- Lerina Aversano analyzed the data, authored or reviewed drafts of the article, and approved the final draft.
- Mario Luca Bernardi conceived and designed the experiments, authored or reviewed drafts of the article, and approved the final draft.
- Marta Cimitile analyzed the data, authored or reviewed drafts of the article, and approved the final draft.
- Andrea Maiellaro conceived and designed the experiments, performed the experiments, analyzed the data, prepared figures and/or tables, authored or reviewed drafts of the article, and approved the final draft.
- Riccardo Pecori conceived and designed the experiments, performed the experiments, analyzed the data, prepared figures and/or tables, authored or reviewed drafts of the article, and approved the final draft.

### Data Availability

This is a literature review.

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
