# Peer review of "A systematic review on artificial intelligence techniques for detecting thyroid diseases"

_PeerJ Computer Science, doi:10.7717/peerj-cs.1394_

## Round 0.1 · original submission · Major Revisions

After several months trying to assign the paper, finally four different reviewers have checked your research work and provided useful comments for improving both the content and the presentation. Some major issues need to be addressed prior publication, especially regarding to the final summary provided in order to provide insights from the topic.

Reviewer 1 ·

Basic reporting

The present study aims to do a systematic review on AI techniques for the thyroid diseases prediction domain. The work fulfills the aims and scope of the journal, and it might be of interest to the machine learning and medicine communities. The work is interdisciplinary and sufficiently broad. The introduction is adequately and accesible to different audience, and the motivation is clear. Here are some comments at the basic reporting:

- The English is clear. Minor language checking is required.
- The literature references are sufficient.
- Figure 1 is quite difficult to understand. Specifically, Figure 1, the green blocks are not clear, try to explain them in a better way.
- From my point of view, Figure 2 does not represent percentage values, but frequencies. Please be clear and modify if needed.
- I suggest to represent Figure 3 and Figure 6 as pie charts.
- Explain the reasoning behind the classification criteria in Table 3.

Experimental design

The work fulfills the aims and scope of the journal, it states a rigorous investigation through a systematic review of the state-of-the-art, and no ethical concerns were found. The method is described sufficiently. Here are some comments to improve the work:

- I recommend to describe the related work with more strong narrative skills. The current text seems to be a list of works, not glueing each other. It would be better to group the works by some similar feature and then provide concise paragraphs reflecting the advantages and limitations of the reviewed works.

Validity of the findings

Some comments on the findings:

- I recommend to include a graph of the taxonomy of the AI techniques in this domain, as a summary of the systematic review. This would beneficial to the readers in terms of visual findings and classification of the state-of-the-art in this particular context.

- It does not seem to be percentages in Figure 4. Rewrite line 250 specifying that numbers are frequencies.

- Specify what do the authors mean with "clinical" in line 514? Does this adjective refer to "tabular data"? Explain and modify if necessary.

- Discussion is quite simple. It really needs to transmit the overall findings in the study. For example, are the AI techniques working or is still in progress, these ML and DL techniques are implemented in laboratory or in real-world, they are using in what contexts (prevention, monitoring, diagnosis, tracking, etc.), it is more valuable to use supervised vs other learning approaches, etc. What are the limits of the AI, what are the limits of the datasets? What would be the opportunities in this particular field. And more importantly, what are the trends on this topic? Is the state-of-the-art focusing on a strategic path to follow up the use of AI in this domain?

- In line 535, what is the optimal value that the authors refer to?

- In line 540, authors refer to feature selection and feature reduction methods, but there are no much information about this topic in the literature review. Include a detailed description of this information in the review.

Reviewer 2 ·

Basic reporting

I examined the study titled “A systematic review on artificial intelligence techniques for predicting thyroid diseases” in detail.

This study aimed to systematically review and analyze the research environment on various techniques related to artificial intelligence applied to the detection and identification of various diseases related to the thyroid gland. In the study, researchers examined 74 articles. In these articles, which methods were used and the performance metrics obtained were examined. They stated that the biggest shortcoming of the studies conducted was the use of old data sets. The points that I was concerned about in the study were presented in articles. It would be more appropriate to include deep learning, CNN, and LSTM-based models in the examined studies. In recent years, deep learning-based methods have been used frequently. Which dataset is used in which study and the characteristics of these datasets should also be discussed. In the study, sentences starting with lowercase letters should be corrected. Apart from these shortcomings, the work is generally well-written.

Experimental design

The work is generally well designed. However, very few articles are presented in Chapter 3.

Validity of the findings

More statistical information from 74 studies is presented. The data used in these studies and the results obtained should also be included.

·

Basic reporting

English needs to improve a lot. Please refer to the attached pdf with comments.
The review is of interest and talks about the different AI technologies used for thyroid detection.
Literature is not written in context at some places. For example in Section 3 Related work, para starting at line 37. The importance of that para is not immediately clear.
The review is important, but authors need to discuss on why THYROID only? That justification seems to be missing in the introduction part.

The results (figures, graphs) need to be properly labelled for y-axis and proper caption.

Experimental design

Study Design is ok. Please Provide PRISMA diagram.
Sources are cited properly.
The review should use smaller, more readable paragraphs.

Validity of the findings

The article seems more like a scoping review. The PICO tool which focuses on the Population, Intervention, Comparison and Outcomes of a (usually quantitative) article, is required if this is a systematic review.

Conclusion is not acceptable in current form. Please refer to detailed comments in the pdf.

Additional comments

Please read the comments in the attached pdf.

Reviewer 4 ·

Basic reporting

This work presents a review of AI based techniques used in prediction of thyroid diseases.

The authors have selected and studied 74 articles to analyse them to answer three main research questions which are
a) which diseases of the thyroid gland are detected,
b) which data set has been used and
c) what types of data are used to perform the detection.

The paper concludes noting that the mostly supervised learning methods are used through private and outdated datasets.

Experimental design

No comment

Validity of the findings

The authors have reported a vast set of literature that uses artificial intelligence for prediction of thyroid diseases
which is limited to hypo-thyroid, hyper-thyroid and thyroid cancer.

They have categorised the reported methods based on the underlying techniques and the data set used.

The reported conclusions don't seem interesting or particularly useful. More importantly, the authors do not include a thorough discussion on advantages and disadvantages of each method. The comparison among data sets and features in each dataset are not included, either. Neither do the discussion and conclusion report any new findings from the literature study.

Additional comments

No comment

---

## Round 0.2 · accepted · Accept

The current version of the manuscript has improved and therefore it is ready for publication.

Reviewer 2 ·

Basic reporting

The authors have corrected the deficiencies in the revision.

Experimental design

ok

Validity of the findings

ok

Additional comments

ok